# Evidence for singular-phonon-induced nematic superconductivity in a topological superconductor candidate $Sr_{0.1}Bi_2Se_3$

Jinghui Wang[1], Kejing Ran[1], Shichao Li[1], Zhen Ma[1], Song Bao[1], Zhengwei Cai[1], Youtian Zhang[1], Kenji Nakajima [2], Seiko Ohira-Kawamura[2], P. Čermák[3,4], A. Schneidewind [3], Sergey Y. Savrasov[5], Xiangang Wan[1,6] & Jinsheng Wen [1,6]

Superconductivity mediated by phonons is typically conventional, exhibiting a momentum-independent $s$-wave pairing function, due to the isotropic interactions between electrons and phonons along different crystalline directions. Here, by performing inelastic neutron scattering measurements on a superconducting single crystal of $Sr_{0.1}Bi_2Se_3$, a prime candidate for realizing topological superconductivity by doping the topological insulator $Bi_2Se_3$, we find that there exist highly anisotropic phonons, with the linewidths of the acoustic phonons increasing substantially at long wavelengths, but only for those along the [001] direction. This observation indicates a large and singular electron-phonon coupling at small momenta, which we propose to give rise to the exotic $p$-wave nematic superconducting pairing in the $M_xBi_2Se_3$ (M = Cu, Sr, Nb) superconductor family. Therefore, we show these superconductors to be example systems where electron-phonon interaction can induce more exotic superconducting pairing than the $s$-wave, consistent with the topological superconductivity.

[1] National Laboratory of Solid State Microstructures and Department of Physics, Nanjing University, Nanjing 210093, China. [2] J-PARC Center, Japan Atomic Energy Agency, Tokai, Ibaraki 319-1195, Japan. [3] Jülich Centre for Neutron Science (JCNS) at Heinz Maier-Leibnitz Zentrum (MLZ), Forschungszentrum Jülich GmbH, Lichtenbergstr. 1, 85748 Garching, Germany. [4] Department of Condensed Matter Physics, Faculty of Mathematics and Physics Charles University, Ke Karlovu 5, 121 16 Praha, Czech Republic. [5] Department of Physics, University of California, Davis, CA 95616, USA. [6] Collaborative Innovation Center of Advanced Microstructures, Nanjing University, Nanjing 210093, China. Correspondence and requests for materials should be addressed to X.W. (email: xgwan@nju.edu.cn) or to J.W. (email: jwen@nju.edu.cn)

For superconductors, the central issue is what drives the otherwise repulsive electrons into bound pairs, which collectively condense below the superconducting transition temperature $T_c$. In conventional superconductors, it is known that the elementary excitations of lattice, phonons, couple with electrons, and act as attractive force that pairs the electrons, resulting in an isotropic $s$-wave superconducting pairing symmetry[1]. The reason behind this is that the electron–phonon interaction is often nearly momentum independent. There are some cases where phonons may play some role in the unconventional superconductivity, for example, in $YBa_2Cu_3O_{7-x}$, but how the electron-phonon coupling is related to the presumable $d$-wave pairing is not clear at the moment[2–5]. Therefore, one natural approach to have a non-$s$-wave superconducting pairing, $d$- or $p$-wave for instance, is to resort to other interactions such as magnetic couplings[6,7]. Another possible route is to seek for strongly momentum dependent electron-phonon coupling[8–11], but extensive research in three-dimensional materials with weak spin-orbit coupling (SOC) has not been quite successful so far.

Recently, the debate on the possibility of realizing topological-insulator-derived topological superconductivity where the electron–electron correlation is weak and the electron-phonon interaction dominates the superconducting pairing, has made this fundamental question more outstanding[12,13]. In this regard, a promising topological superconductor candidate $Sr_{0.1}Bi_2Se_3$ with quasi-two-dimensional structure and strong SOC has come to our attention. $Sr_{0.1}Bi_2Se_3$ is a member of the $M_xBi_2Se_3$ (M=Cu, Sr, Nb) family, which become superconducting by doping the topological insulator $Bi_2Se_3$[14–38]. $Bi_2Se_3$ crystallises in a layered hexagonal structure with the space group $R\bar{3}m$ (Fig. 1a). Superconductivity in this family was initially discovered in $Cu_xBi_2Se_3$ by intercalating Cu into the gaps between the Se layers of $Bi_2Se_3$[16,17], and subsequently in $Sr_xBi_2Se_3$ and $Nb_xBi_2Se_3$, with the latter two having larger superconducting volume fractions[21–23]. Experimental studies from nuclear magnetic resonance[26], specific heat[27,28], penetration depth[29], transport[30,31], and magnetic torque[32], all indicate that the superconducting order parameter breaks the in-plane three-fold crystalline rotation symmetry, and exhibits a two-fold nematic pattern consistent with that of a $p$-wave pairing.

A number of theoretical works have also proposed the exotic $p$-wave superconductivity in this family[18,33–38]. Based on a short-range electron density–density interaction in a two-orbital model, Fu and Berg proposed a spin-triplet pairing with odd parity in Cu-doped $Bi_2Se_3$[18]. The presence of strong SOC and hexagonal warping in $Bi_2Se_3$ had been suggested to play an important role in rendering the unconventional superconducting behaviours[33,34]. By performing symmetry analyses, Yang et al. also pointed out the possibility of realizing a $p$-wave superconductivity in $Cu_xBi_2Se_3$[35]. For such a system with $p$ electrons where electronic correlations are weak, it is most likely that the electron-phonon interaction will be the main driving force for the superconductivity. Theoretically, it is indeed proposed that such an non-$s$-wave superconducting pairing in $Cu_xBi_2Se_3$ is driven by the electron–phonon interaction, which is usually expected to result in a conventional $s$-wave superconductivity[36,37]. In particular, based on first-principles calculations, a singular electron–phonon interaction at long wavelength had been suggested in $Cu_xBi_2Se_3$[36]. This unusual electron–phonon interaction was shown to make

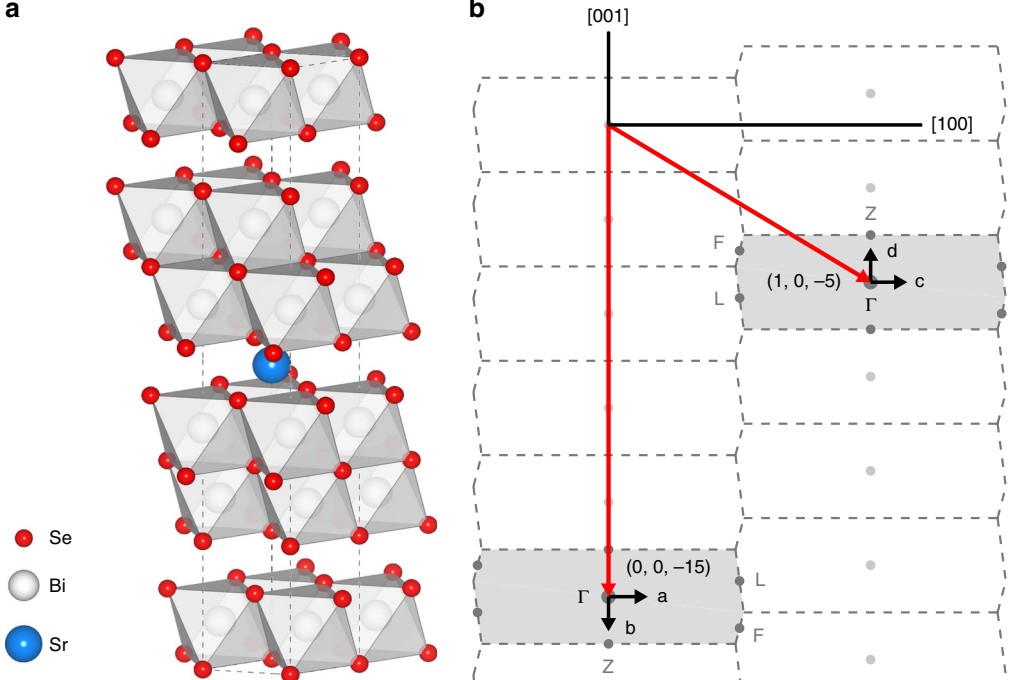

**Fig. 1** Schematic crystal structure and the experimental scheme of the neutron scattering measurements. **a** Schematic of the hexagonal crystal structure of $Sr_{0.1}Bi_2Se_3$. The dopant Sr ions are believed to be intercalated in the van der Waals gaps between the quintuple layers consisting of Bi and Se ions[39]. The dashed lines denote a unit cell, while the shades denote the octahedra formed by the Se ions, with Bi ions in the centre. **b** Brillouin zones in the (H0L) plane defined by two orthogonal axes [100] and [001] in the reciprocal space, constructed based on the hexagonal notation illustrated in (**a**). The two long arrows point to (1, 0, −5) and (0, 0, −15), the two strongest Bragg peaks available around which we measure the phonons. Arrows on (1, 0, −5) and (0, 0, −15) represent the directions along which we plot the phonon dispersions in Fig. 2. The letters a–d correspond to panels **a**–**d** in Fig. 2. Dashed lines illustrate the Brillouin zone boundaries, and the shades represent the two measured Brillouin zones. Dots represent high-symmetry points, with Γ, Z, F, and L in the measured Brillouin zones being labelled. The wave vector **Q** is expressed as (H, K, L) reciprocal lattice unit (rlu) of $(a^*, b^*, c^*) = (4\pi/\sqrt{3}a, 4\pi/\sqrt{3}b, 2\pi/c)$, with $a = b = 4.14$ Å, and $c = 28.6$ Å

different pairing channels have similar strength, and the $A_{2u}$ symmetric $p_z$ pairing should win if the effect of the Coulomb pseudopotential $\mu^*$ was taken into account[36]. Similarly, Brydon et al. also proposed an odd-parity superconductivity in the same material by considering the combined effect of the electron–phonon interaction and $\mu^{*}$[37]. Based on an effective model for $Bi_2Se_3$, the on-site repulsive interaction had been carefully studied, and the results showed that the Coulomb interaction generated repulsive interaction in both the $s$-wave and $A_{2u}$ paring channels, but not in the $E_u$ ($p_x$ or $p_y$ pairing) channel, making the latter win over the former two[38]. Therefore, examining the electron-phonon interaction experimentally is the key to the understanding of the superconducting pairing mechanism in the $M_xBi_2Se_3$ (M=Cu, Sr, Nb) superconductor family.

Here, using inelastic neutron scattering (INS), we measure the phonons on a superconducting single crystal of $Sr_{0.1}Bi_2Se_3$, isostructural to $Cu_xBi_2Se_3$ but with a larger superconducting volume fraction[21,22]. We show that the phonon linewidths of the acoustic mode measured along the [001] direction increase significantly when approaching the Brillouin zone centre. This observation reflects the singular electron-phonon pairing interaction which diverges at small momenta ($q$s), and may be responsible for the exotic $p$-wave superconducting pairing symmetry discussed extensively[18,26,27,29–38].

## Results

**Phonon dispersions**. The $Sr_{0.1}Bi_2Se_3$ single crystals with a $T_c = 3.2\,K$ used in this work have been well characterised as described in the Supplementary Fig. 1 and in ref. [39]. We have carried out INS measurements on the single crystal using an experimental scheme sketched in Fig. 1b. We focus on two accessibly strongest Bragg peaks (0, 0, −15) and (1, 0, −5) and map out the phonons at low energies around these peaks. At (0, 0, −15), as the wave vector **Q** is parallel to the [001] direction, phonons propagating along [001] and [100] directions are purely longitudinal and transverse modes, respectively. On the other hand, at (1, 0 −5), along both [001] and [100] directions, there is a mix of the longitudinal and transverse phonon modes, as these directions are in between the longitudinal and transverse directions, as illustrated in Fig. 1b.

In Fig. 2, we show the phonon dispersions around (0, 0, −15) and (1, 0, −5) for $Sr_{0.1}Bi_2Se_3$ measured at $T = 17\,K$. These low-energy branches correspond to the motions of the heavier Bi atoms[36]. Around (0, 0, −15), we observe one dispersing transverse acoustic (TA) mode up to ~5 meV along the [100] direction (Fig. 2a); along the [001] direction, a softer longitudinal acoustic (LA) mode is observed with a band top of ~3 meV (Fig. 2b). Around (1, 0, −5), we find one TA and one LA mode along the [100] direction (Fig. 2c); along the [001] direction, besides the softer TA and LA modes, we also observe an almost dispersionless transverse optic (TO) mode (Fig. 2d). The TO mode at $q = 0.4\,rlu$ has an energy of ~4.85 meV, which is very close to the zone centre optic mode at 4.82 meV obtained from Raman scattering[40]. Overall, these data are in reasonable agreement with previous numerical results[36,40–42], with phonons somewhat softer than those calculated in ref. [36].

We have also examined the phonons at $T = 0.8\,K$, below the $T_c$ of 3.2 K, but do not observe any essential changes on the phonon spectra across the $T_c$. Therefore, in the rest of the paper, we will discuss our results obtained at $T = 17\,K$. We note that for some superconductors with a higher $T_c$, there have been some reports showing changes for certain phonon modes matching twice of the superconducting gap $2\Delta(T)$ across $T_c$[43–46], likely due to the participation in the electron-phonon coupling for these modes[47].

**Linewidth broadenings near the zone centre**. We perform energy cuts at a series of $q$s on the phonon dispersions shown in Fig. 2, and the results of the cuts are plotted in Fig. 3. From the cuts, we see that the phonon linewidth at different $q$s along the [100] direction for the two TA modes around $(q, 0, −15)$ and $(1 + q, 0, −5)$, shown in Fig. 3a, c, respectively, do not show noticeable changes. Similarly, as shown in Fig. 3d, around (1, 0, −5), along the [001] direction, the linewidth of the LA and TO modes has no $q$ dependence. On the other hand, for the LA and TA phonons along the [001] direction, shown in Fig. 3b, d, respectively, we do clearly see that they become much broader when approaching the zone centre. For instance, in Fig. 3b, the phonon is sharp and almost resolution limited at large $q$s such as at $q = 1.0–1.3\,rlu$. When $q = 0.5\,rlu$, the phonon becomes very broad. As can be clearly seen from Fig. 3d, the TA mode becomes significantly broader for $q \lesssim 1.0\,rlu$, well before the energy of the LA mode getting too close to be resolved from the TA mode. In other words, the linewidth increase observed in Fig. 3d is not due to the merge of the TA and LA modes.

To better characterise the phonon linewidth, we have fitted the energy scans in Fig. 3 using Lorentzian functions convoluted with the instrumental resolution, and plotted the fitted width as a function of $q$ in Fig. 4. The results show that along the [100] direction, the linewidth remains almost constant for the TA modes around (0, 0, −15) and (1, 0, −5) (Fig. 4a); by contrast, as shown in Fig. 4b, along the [001] direction, the width increases from 0.15 meV at $q = 1.3\,rlu$ to 1.0 meV at $q = 0.5\,rlu$. For comparison purpose, we have performed similar measurements on the phonons of the undoped compound $Bi_2Se_3$, and the obtained phonon dispersions are shown in Supplementary Fig. 2. In Fig. 4, we also plot the phonon linewidth as a function of $q$ for $Bi_2Se_3$. The results are very similar to those of $Sr_{0.1}Bi_2Se_3$, where the low-energy phonons only become broader along the [001] direction at small $q$s. However, compared to $Sr_{0.1}Bi_2Se_3$, the linewidth increase is much reduced, indicating that Sr doping can enhance the broadening effect at small $q$s. The evolution of the phonon linewidth has actually been predicted in an isostructural compound $Cu_xBi_2Se_3$ in ref. [36], where it is shown that the acoustic phonons remain well defined at large $q$s along the [001] direction or in the whole $q$ range along other directions. In other words, according to the calculations[36], the acoustic phonons will only show broadenings at small $q$s along the [001] direction, which is exactly what we show in Fig. 4.

## Discussions

By far, we have demonstrated that the acoustic phonons become broad only along the [001] direction at small $q$s, consistent with the theoretical calculations[36]. What is the origin of such singular phonons and what is the consequence? Since $Bi_2Se_3$ has both time-reversal and spatial-inversion symmetry, every energy band is at least double degenerate. With the strong SOC, the phonon displacement along the [001] direction at small $q$s which breaks the spatial-inversion symmetry efficiently lifts the double degeneracy, resulting in a large and singular electron-phonon coupling matrix element along this direction near the zone centre. Furthermore, because of an open-cylinder-like electron pocket along the Γ-Z direction centering at the Γ point[48], it is shown that there is a Fermi surface nesting along this direction, and the nesting function has the largest value as $q \rightarrow 0$[36]. The combining effect of the large and singular electron-phonon interaction as well as the strong Fermi surface nesting gives rise to the large linewidth for the phonons along the [001] direction at small $q$s. The contribution of each phonon mode to the electron-phonon coupling constant is proportional to the phonon linewidth divided by the square of the phonon energy. Therefore, the

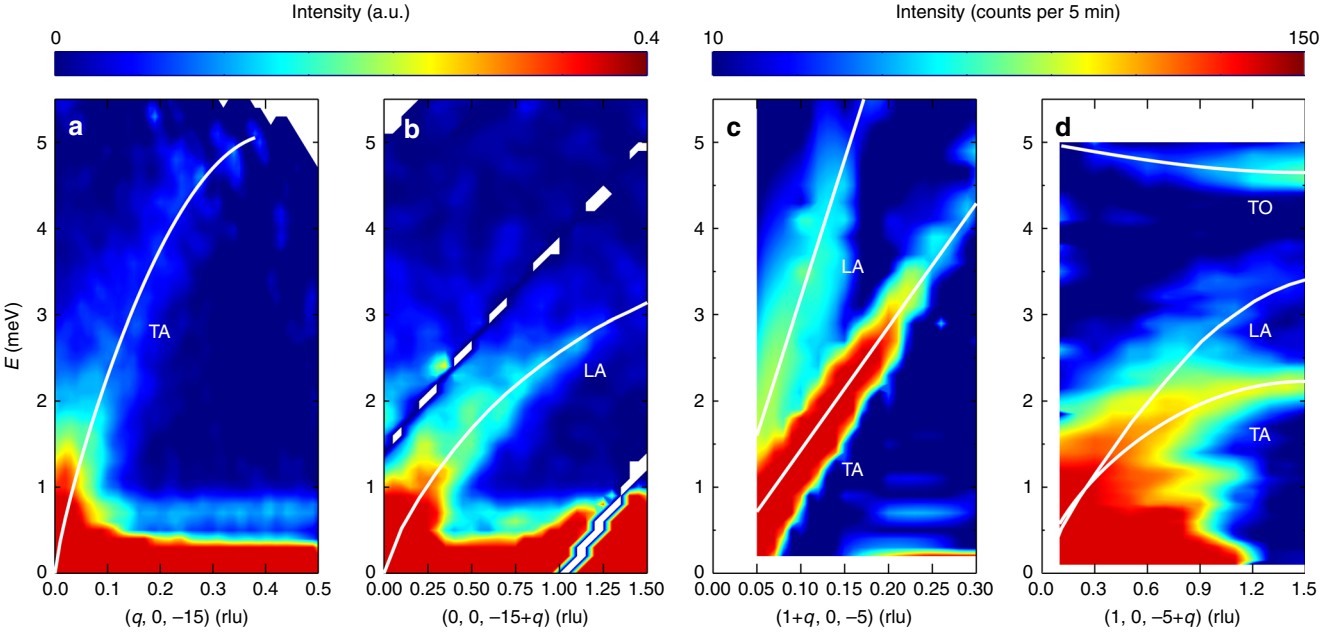

**Fig. 2** Phonon dispersions around two Bragg peaks (0, 0, −15) and (1, 0, −5) measured at $T = 17$ K. **a, b** phonon dispersing from (0, 0, −15) along the [100] and [001] directions, respectively. **c, d**, phonon dispersing from (1, 0, −5) along the [100] and [001] directions, respectively. The data plotted in **a, b** were collected on a time-of-flight (TOF) spectrometer AMATERAS, and those in **c, d** were collected on a triple-axis spectrometer (TAS) PANDA. For the TOF data plotted in **a, b**, which are the phonon dispersions along one direction, one needs to sum over intensities for a certain thickness along the other two directions to improve the statistics. The data plotted against the $H$ direction in **a** were obtained by integrating the intensities in ($q$, $0 + K$, $-15 + L$) with a thickness of $K$ and $L$ ranging from −0.1 to 0.1 rlu, and −16 to −14 rlu, respectively. Those in **b** were integrated with $H = K = [−0.05, 0.05]$ rlu. $E$ and $q$ represent the phonon energy and momentum, respectively. TA, LA, and TO represent transverse acoustic, longitudinal acoustic and transverse optic modes, respectively. Solid lines are the dispersions obtained by fitting the energy scans at a series of $q$s as shown in Fig. 3. The white streaks along the diagonal direction in **b** were due to the lack of detector coverage wherein

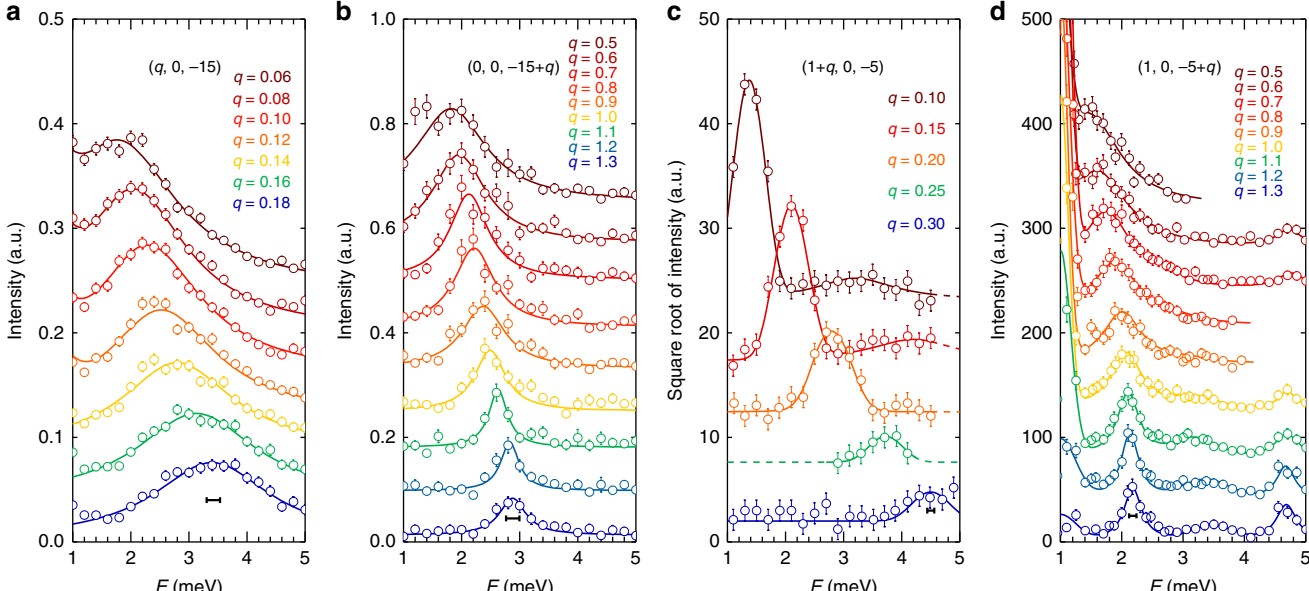

**Fig. 3** Energy cuts on the phonon dispersions. **a–d** Correspond to the energy cuts on the dispersions plotted in Fig. 2a–d at different $q$ values. The cuts are offset so that each cut can be visualized clearly. In **c** due to the large intensity difference of phonons at small and large $q$s, we plot the square root of the intensities as a function of energy so that the cut profiles at large $q$s can be visible. Solid lines are fits with Lorentzian functions convoluted the instrumental resolutions, as indicated by the horizontal bars. From the fits, we obtain the phonon dispersions, which are plotted as solid lines in Fig. 2. Errors represent one standard deviation throughout the paper

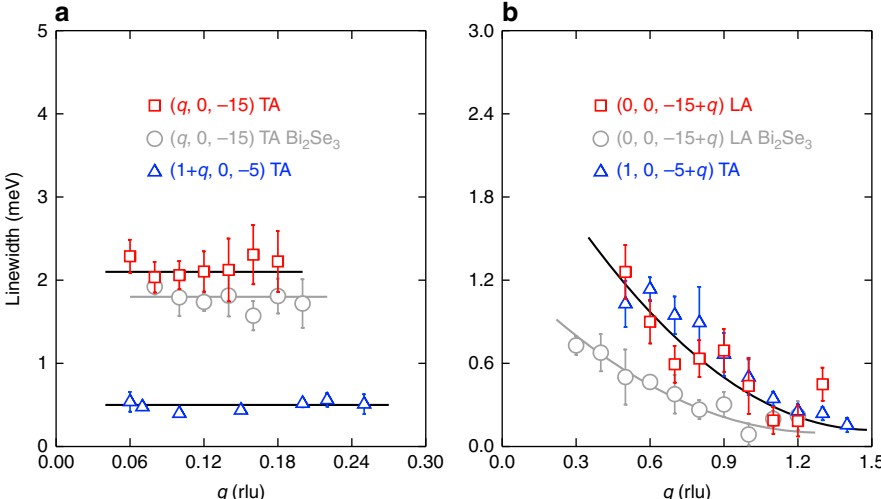

**Fig. 4** Evolution of the phonon linewidth with $q$. **a, b** Phonons along the [100] and [001] directions, respectively. Squares and triangles represent results extracted from data collected on a TOF spectrometer AMATERAS and TAS PANDA, respectively. For comparison, the linewidths of the $Bi_2Se_3$ phonons measured on AMATERAS are also plotted. Lines through data are guides to eye to illustrate the evolution of the phonon linewidth as a function of $q$

broad acoustic phonons at small $q$s observed here should dominate the electron-phonon coupling. As shown in Fig. 4b, the linewidth increase is larger for $Sr_{0.1}Bi_2Se_3$ than that for the $Bi_2Se_3$ sample. In this sense, the Sr doping not only brings in the electrons[16,39], but also enhances the electron-phonon coupling, both of which drive the superconductivity in $Sr_{0.1}Bi_2Se_3$.

Based on a symmetry analysis, one can expect that this highly unusual electron-phonon interaction results in a remarkable proximity of all pairing channels. The strengths of the $s$- and $p$-wave pairing interaction are indeed shown to be comparable in this case[36]. Naively one may expect that the Coulomb interaction $\mu^\star$ only suppresses the $s$-wave pairing and a moderate $\mu^\star$ should result in an $A_{2u}$ ($p_z$) symmetric pairing[36]. However, a realistic calculation reveals that the Coulomb interaction in $Bi_2Se_3$ generates repulsive interaction in both the $s$-wave and $A_{2u}$ pairing channels but not in the $E_u$ ($p_x$ or $p_y$) channel[38]. Considering that the strength of the $E_u$ pairing is comparable to that of the $s$-wave pairing from first-principles linear response calculations, one can thus expect a two-fold nematic $E_u$ pairing which breaks the in-plane three-fold crystalline rotation symmetry in this system[36,38]. Alternatively, Fu and Berg suggest that the $E_u$ pairing states can also be realized as a consequence of the inter-orbital pairing, and the broadening of the [001] phonons observed here may assist such inter-orbital pairing[18].

As we show in Supplementary Fig. 3, for the superconducting sample $Sr_{0.1}Bi_2Se_3$ we study here, both the resistance and upper critical field exhibit a clear two-fold pattern, consistent with previous experimental reports on the nematic $p$-wave superconductivity in the $M_xBi_2Se_3$ (M=Cu, Sr, Nb) superconductor family[26,27,29–32]. Our work thus provides concrete evidence that an odd-parity $p$-wave pairing compatible with that of the topological superconductivity can be induced by strongly anisotropic electron-phonon interaction. Very recently, thousands of topological materials based on weakly-correlated systems have been proposed theoretically[49–51]. Since we show here that topological superconductivity can exist in such systems, the search for topological superconductors should be much more promising.

## Methods

**Single-crystal growth and characterisations**. High-quality single crystals of $Bi_2Se_3$ and $Sr_{0.1}Bi_2Se_3$ were grown using the horizontal Bridgman method. The stoichiometric raw materials (99.99% Sr, 99.99% Bi and 99.99% Se powders) of nominal composition were mixed well and sealed in an ampoule in vacuum. The

ampoule was then placed in a tube furnace with a temperature gradient, heated up to 850 °C, stayed for 2 days, and cooled from 850 to 610 °C at 1 °C per hour for the melt to crystallise. The resistance and magnetization were measured in a physical property measurement system (PPMS-9T, Quantum Design). The PPMS was equipped with a sample rotator used to measure the angular dependence of the electrical properties.

**INS experiments**. INS measurements were performed on the cold triple-axis spectrometer (TAS) PANDA located at FRM-II, Germany, and cold-neutron time-of-flight (TOF) disk-chopper spectrometer AMATERAS located at J-PARC, Tokai, Ibaraki, Japan. On the TAS, we used a fixed final energy $E_f$ mode. For the experiment on PANDA, one single crystal of $Sr_{0.1}Bi_2Se_3$ with a mass of 1.1 g was mounted in the $(H, 0, L)$ scattering plane. The sample mosaic was about 5 degrees (FWHM, full width at half maximum) scanned around the Bragg peak (1, 0, −5). Double-focusing mode was used for both the monochromator and analyser. The collimation is 80′-80′-sample-80′-80′. With these configurations, the energy resolutions were about 0.09 meV for $E_f = 5.1$ meV and 0.05 meV for $E_f = 3.5$ meV. To reduce higher-order neutrons, one Be filter was placed after the sample. A closed-cycle refrigerator (CCR) equipped with a $^3$He insert was used to reach the base temperature of 0.8 K. In the TAS experiments, we measured the phonon dispersion along the direction that matched the resolution ellipse to improve the resolution. On the TOF spectrometer AMATERAS, a multiple-$E_i$ (incident energy) mode with main $E_i = 10.5$ meV and resolution $\Delta E = 0.33$ meV at $E = 0$ meV was used. For $Sr_{0.1}Bi_2Se_3$, the 1.1-g sample measured on PANDA was used on AMATERAS. We also measured a $Bi_2Se_3$ sample with a larger mass of 1.6 g for comparison. Both samples were mounted with $c$ axis along the beam direction and $a$ axis in the horizontal plane. The rotation axis was perpendicular to $a$-$c$ plane, and the step was 0.5 degree around the Bragg peaks (0, 0, −15) and (1, 0, −5). A $^4$He CCR with a temperature range of 5–300 K was used on AMATERAS. The wave vector $\mathbf{Q}$ is expressed as $(H, K, L)$ reciprocal lattice unit (rlu) of $(a^*, b^*, c^*) = (4\pi/\sqrt{3}a, 4\pi/\sqrt{3}b, 2\pi/c)$, with $a = b = 4.14$ Å and $c = 28.6$ Å for both $Bi_2Se_3$ and $Sr_{0.1}Bi_2Se_3$.

## Data availability

Data are available from J.S.W. (Email: jwen@nju.edu.cn) upon reasonable request. The source data underlying Figs. 2–4 and Supplementary Figs 1–3 are provided as a Source Data file.

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

## Acknowledgements

The work was supported by the National Natural Science Foundation of China with Grant Nos 11822405, 11674157, 11525417, and 11834006, NSF Jiangsu province in China with Grant No. BK20180006, Fundamental Research Funds for the Central Universities with Grant No. 020414380117, and the Office of International Cooperation and Exchanges of Nanjing University. S.Y.S. was supported by NSF DMR with Grant No. 1832728. We thank Shengyuan Yang, Fan Zhang, Haijun Zhang, Hai-Hu Wen, and Qiang-Hua Wang for stimulating discussions.

## Author contributions

J.S.W. and X.G.W. conceived the project. J.H.W. grew and characterised the single crystals with assistance from K.J.R., S.C.L., Z.M., S.B., Z.W.C. and Y.T.Z. J.S.W. and J.H. W. carried out the neutron scattering experiments with help from K.N., S.O.K., P.C. and A.S. J.H.W. and J.S.W. analysed the data. J.S.W., X.G.W. and J.H.W. wrote the paper with inputs and comments from S.Y.S. and all other authors.

## Additional information

**Competing interests:** The authors declare no competing interests.

