## [Peer Review File · Nature Communications]

Reviewers' comments:

Reviewer #1 (Remarks to the Author):

(0) I have reviewed the manuscript NCOMMS-19-04391A-Z "Evidence for singular-phonon-induced nematic superconductivity in a topological superconductor candidate $\text{Sr}_{0.1}\text{Bi}_2\text{Se}_3$ " submitted for publication in Nature Communications.

In this manuscript, the authors report phonon dispersions of $\text{Sr}_{0.1}\text{Bi}_2\text{Se}_3$ using the inelastic neutron scattering (INS) technique. This compound, as well as its sister compound with different dopants, is attracting much attention as a promising candidate for topological and nematic superconductors. The authors found that the phonons along the [001] direction exhibits significant broadening as the wavenumber approaches the zone center, indicating that the electron-phonon interaction is singularly strong for these phonon branches. Considering existing theories, this result might provide evidence for phonon-mediated unconventional and nematic superconductivity in the doped Bi_2Se_3 systems.

The observation reported in the manuscript is quite interesting, and the possible evidence for phonon-mediated unconventional pairing can attract much attention from wide audience in the materials science community. I expect that this report stimulates significantly the study of unconventional superconductivity in topological materials. Nevertheless, I raise several points that can be improved. In particular, the present analysis and discussion is rather qualitative and should be made more quantitative.

To summarize, the manuscript is worth for publication in Nature Communications but after a suitable revision is made.

Below, I list my detailed point-by-point comments that should be addressed on resubmission.

(1) In this manuscript, there are so many "guide to eye". Some of them should be replaced with more quantitative curves.

(a) In Fig.2, the phonon dispersions given are just explained as "guides to eye". Explain how they are actually determined? Fitting to the peak position or theoretical calculation? It would be nicer to compare the experimental data with the theoretical calculation (Ref. [23]), whose authors are actually among the present authors. Then, add some discussion on the agreement/difference between the experiment and calculation. Similar improvements should be done also for Supple. Fig. 2.

(b) Also, the "guides to eye" curves in Fig.4 should be replaced with some fitting or with theoretical curves. Since the broadening of the phonon spectra is predicted in Ref. [23], it would be not so difficult to compare theoretical and experimental broadenings and to add discussion on their similarities/differences.

(2) Comparison of the present observation with other experiments is missing.

(a) Compare the present results with those observed in other unconventional superconductors. In particular, in cuprate, there are discussions that the phonons may be playing important roles. Such comparison is quite important to claim "the first examples where electron-phonon interaction can induce more exotic superconducting pairing than the s-wave".

(b) Compare the present results with the phonon information on the Bi_2Se_3 systems investigated with other techniques, e.g. Raman spectroscopy in Gnezdilov et al. [PRB 84, 195118 (2011).]

(3) The mechanism of the phonon-driven nematic superconductivity is not clearly explained.

(a) In Ref.[23], the "odd-parity" fluctuation is important. Explain whether the phonon modes with the significant broadening are actually accompanied by such "odd-parity fluctuation".

(b) It would be better to mention that the Eu pairing states are realized as a consequence of the inter-orbital (i.e. inter-layer) pairing (See Ref.[5] etc.) and that the broadening of the [001] phonons may assist such inter-orbital pairing.

(4) Some important experimental conditions and details are not explained. Explanation of such information is particularly important for the broad audience of Nature Communications, but also for specialists.

(a) At what temperature data shown in each figure was taken? Were there any change between the phonon spectra below and above T_c ?

(b) Are the [100] axis (along which the phonon dispersion was taken) and the nematic superconducting axis (one of the [100] axes where H_{c2} exhibit maximum) common or not? If the authors had measured the dispersion both along the [100] and [010], comment whether nematicity is found in the phonon dispersion or not.

(c) The explanation in the figure captions such as "integrated with $K = [-0.1, 0.1]$ rlu and $L = [-16, -14]$ rlu" is not clear to non-specialists. Some more explanation should be added about such integration process.

(d) What are the white regions extending diagonally in Fig. 2b and Supple. Fig. 2b? I guess that data is missing because of the configuration of the spectrometer. But please explain.

(e) In Fig. 3, the data are offset vertically. Explain how much offset is added.

(5) Comments on the presentations of the data.

(a) For Fig. 2, it would be better if the vertical scale is common in the all four panels. i.e. all data should be plotted in the energy range between 0 and 6 meV. In addition, for Fig. 2c and d, the horizontal range should include $q = 0$. If data outside of the present ranges is missing, such regions can be left white.

(b) For Fig. 3c, it is written in the figure caption that square root of intensity is plotted. Then, the label of the vertical axis should be modified.

(c) In Fig. 3d, explain how the dashed curves (extrapolating the spectra for $q = 0.5, 0.8,$ and 0.9 to 5 meV) are drawn. If there is no good reason to estimate the position and shape of the TO peaks even without measurements, these dashed curves should be removed since they can be misleading.

(6) Other minor comments.

(a) In the "Author Contributions", some authors are not mentioned.

Reviewer #2 (Remarks to the Author):

This is a very interesting paper that should be published after the authors have addressed the following minor points:

1) The in-plane electrical, thermal, NMR and penetration depth studies indicate a two-fold symmetry rather than the 3 fold symmetry of the crystal structure. This nematic behavior is consistent with p wave superconductivity but may not be definitive test. Do the authors have any knowledge of how the Sr dopants are spatially arranged between the layers?

2.) In Fig 2a, why is the TA mode so difficult to see compared with Fig 2a in the supplementary

information?

3.) On page 3 , right column, line 4- it should read "remarkable proximity of all pairing channels." It would be helpful if this statement were expanded a bit using the results from ref 23, since this is the key theoretical result that led to the present experiments.

4.) Supplementary Fig Caption 1: The last statement that a shielding fraction of 95% indicates a large superconducting fraction is not always true. Filamentary superconductivity can produce a similar result. A heat capacity feature near T_c or a large Meisner fraction are better indicators of bulk superconductivity.

5) The paper by Willa et al. on the calorimetric evidence for nematic sc in $\text{Sr}_{0.1}\text{Bi}_2\text{Se}_3$ should be cited. (PRB 98, 184509 (2018)).

Response to the Reviewers

We thank both reviewers for their efforts in reviewing our manuscript and for the positive comments on our work. We are also grateful for both reviewers for providing many valuable suggestions based on which we can improve our work. We have taken all the suggestions into consideration and modified the manuscript accordingly. The changes are marked in red in the manuscript and a summary for changes is appended after our point-by-point response to the reviewers. We hope that both reviewers will find the revised manuscript satisfactory and recommend it for publication in Nature Communications.

Response to Reviewer #1

Comments:

(0) I have reviewed the manuscript NCOMMS-19-04391A-Z "Evidence for singular-phonon-induced nematic superconductivity in a topological superconductor candidate $\text{Sr}_{0.1}\text{Bi}_2\text{Se}_3$ " submitted for publication in Nature Communications.

In this manuscript, the authors report phonon dispersions of $\text{Sr}_{0.1}\text{Bi}_2\text{Se}_3$ using the inelastic neutron scattering (INS) technique. This compound, as well as its sister compound with different dopants, is attracting much attention as a promising candidate for topological and nematic superconductors. The authors found that the phonons along the [001] direction exhibits significant broadening as the wavenumber approaches the zone center, indicating that the electron-phonon interaction is singularly strong for these phonon branches. Considering existing theories, this result might provide evidence for phonon-mediated unconventional and nematic superconductivity in the doped Bi_2Se_3 systems.

The observation reported in the manuscript is quite interesting, and the possible evidence for phonon-mediated unconventional pairing can attract much attention from wide audience in the materials science community. I expect that this report stimulates significantly the study of unconventional superconductivity in topological materials. Nevertheless, I raise several points that can be improved. In particular, the present analysis and discussion is rather qualitative and should be made more quantitative.

To summarize, the manuscript is worth for publication in Nature Communications but after a suitable revision is made.

Below, I list my detailed point-by-point comments that should be addressed on resubmission.

Response:

We thank the reviewer very much for the detailed and insightful appraisal of our work. We are thankful to the reviewer for considering "*The observation reported in the*

manuscript is quite interesting, and the possible evidence for phonon-mediated unconventional pairing can attract much attention from wide audience in the materials science community. I expect that this report stimulates significantly the study of unconventional superconductivity in topological materials.”

We also greatly appreciate the reviewer for raising many points that can substantially improve the work. As we elaborate from the point-by-point response below, we have addressed all these issues and made changes accordingly in the revised manuscript.

Comments:

(1) In this manuscript, there are so many "guide to eye". Some of them should be replaced with more quantitative curves.

(a) In Fig.2, the phonon dispersions given are just explained as "guides to eye". Explain how they are actually determined? Fitting to the peak position or theoretical calculation? It would be nicer to compare the experimental data with the theoretical calculation (Ref. [23]), whose authors are actually among the present authors. Then, add some discussion on the agreement/difference between the experiment and calculation. Similar improvements should be done also for Supple. Fig. 2.

Response:

We thank the reviewer very much for this useful comment. Now, instead of using guide lines to illustrate the phonon dispersions, we obtain the phonon dispersions by connecting the fitted peak positions in energy from fitting the energy scans at a series of reduced wave vectors q s. Some of the scans and fittings are shown in Fig. 3. Similarly, we have obtained the dispersions for Supplementary Fig. 2, and updated both Fig. 2 and Supplementary Fig. 2 with the new dispersions so obtained. We have explained how we obtain the dispersions in the captions.

Overall, the calculated results in Nat Commun. 5, 4144 (2014) (the original Ref. [23]) agree reasonably well with the experimental results, with the calculated phonons being somewhat harder than the experimental ones. As suggested by the reviewer in Comment #(2)(b), we compare our results with the Raman results in [PRB 84, 195118 (2011)], which probed the zone center $q=0$ optic mode in $\text{Cu}_x\text{Bi}_2\text{Se}_3$. They observed an $E_g(1)$ mode with an energy of 4.82 meV, which is nearly doping independent. We observe a nearly dispersionless optic mode with an energy of 4.85 meV at $q=0.4$ rlu. These results are in excellent agreements. In the revised manuscript, we have added the Raman work as Ref. 40, and discussed the comparison between our results and the theoretical [37,40-42] as well as Raman results [40].

(b) Also, the "guides to eye" curves in Fig.4 should be replaced with some fitting or with theoretical curves. Since the broadening of the phonon spectra is predicted in Ref. [23], it would be not so difficult to compare theoretical and experimental broadenings and to add discussion on their similarities/differences.

Response:

We agree with the reviewer that it would be nicer if there could be some theoretically calculated curves to be overlaid on the experimental data points. As illustrated in Ref. [23], the phonon linewidth γ is governed by the Fermi surface nesting function and electron-phonon coupling matrix element. An overall quantitative estimation of γ should ask for a realistic determination for both parts. Even so, the calculation is indeed well consistent with our results qualitatively, which captures the feature that phonons only broaden at small q s along [001] but remain well defined at large q s or in the whole q range along other directions. The purpose of the guide lines is to show these trends, which have been discussed in comparison to the theoretical results in the revised manuscript.

Comments:

- (2) Comparison of the present observation with other experiments is missing.
- (a) Compare the present results with those observed in other unconventional superconductors. In particular, in cuprate, there are discussions that the phonons may be playing important roles. Such comparison is quite important to claim "the first examples where electron-phonon interaction can induce more exotic superconducting pairing than the s-wave".

Response:

We thank the reviewer for this very important point. There are indeed some discussions on other unconventional superconductors, *e.g.*, $\text{YBa}_2\text{Cu}_3\text{O}_{7-x}$ where phonons can play some role [PRB 38, 4992 (1988); PRL 65, 915 (1990); PRL 69, 359 (1992); PRL 70, 1457 (1993)]. However, in these superconductors, what role phonons play, and how electron-phonon coupling is related to the presumable *d*-wave pairing are not quite clear at the moment. We have added these papers as new references 2-5 and made discussions accordingly. Also, to comply with the journal's "no claims of primacy" policy, we have removed the word "first" and rephrased the sentence as "Therefore, we show these superconductors to be example systems where electron-phonon interaction can induce more exotic superconducting pairing than the *s*-wave, consistent with the topological superconductivity."

- (b) Compare the present results with the phonon information on the Bi_2Se_3 systems investigated with other techniques, *e.g.* Raman spectroscopy in Gnezdilov et al. [PRB 84, 195118 (2011).]

Response:

As discussed in the response to Comment #(1)(a), the optic branch obtained from our inelastic neutron scattering experiment is in excellent agreement with the Raman results. This paper is now cited as Ref. 40.

Comments:

- (3) The mechanism of the phonon-driven nematic superconductivity is not clearly

explained.

(a) In Ref.[23], the "odd-parity" fluctuation is important. Explain whether the phonon modes with the significant broadening are actually accompanied by such "odd-parity fluctuation".

Response:

We agree with the reviewer that we should discuss the phonon-driven nematic superconductivity in more details. In Nat Commun. 5, 4144 (2014) (the original Ref. [23]), the proposal is as following: Since Bi_2Se_3 has both time-reversal and spatial-inversion symmetry, every energy band is at least double degenerate. With the strong spin-orbit coupling, the phonon displacement along the [001] direction at small q s which breaks the spatial inversion symmetry efficiently lifts the double degeneracy, resulting in a large and singular electron-phonon coupling matrix element along this direction near the zone centre. Furthermore, because of an open-cylinder-like electron pocket along this direction centring at the zone center [PRB 88, 195107 (2013)], the Fermi surface nesting function is also shown to be the largest as q approaches 0 along the [001] direction. The combining effect of the large and singular electron-phonon interaction as well as the strong Fermi surface nesting gives rise to the large phonon linewidth along the [001] direction at small q s. The contribution of each phonon mode to the electron-phonon coupling constant is proportional to the phonon linewidth divided by the square of the phonon energy. Therefore, the broad acoustic phonons at small q s along the [001] direction should dominate the electron-phonon coupling. Taking all these into account, the pairing strengths of both s - and p -channels are calculated to be comparable. By further considering Coulomb interaction, it is shown that the odd-parity E_u pairing (p_x or p_y) should win [PRB 96, 144504 (2017)]. In the revised manuscript, we have added more discussions to explain the mechanism of the phonon-induced nematic superconductivity as suggested by the reviewer.

(b) It would be better to mention that the E_u pairing states are realized as a consequence of the inter-orbital (i.e. inter-layer) pairing (See Ref.[5] etc.) and that the broadening of the [001] phonons may assist such inter-orbital pairing.

Response:

We thank the reviewer for suggesting this scenario for realizing the E_u pairing. In the revised manuscript, we have discussed the E_u pairing states in the context of phonon-assisted inter-orbital pairing.

Comments:

(4) Some important experimental conditions and details are not explained. Explanation of such information is particularly important for the broad audience of Nature Communications, but also for specialists.

(a) At what temperature data shown in each figure was taken? Were there any change between the phonon spectra below and above T_c ?

Response:

We are sorry for the carelessness. The data were collected at 17 K and the temperature has now been stated explicitly in the manuscript.

As the reviewer may have already noticed, for phonon-driven superconductors, there may be some changes across T_c [PRL 30, 214 (1973); PRB 12, 4899 (1975); PRL 101, 237002 (2008); PRB 82, 024509 (2010)], for certain phonon modes matching twice of the superconducting gap $2\Delta(T)$, due to the participation in the electron-phonon coupling for these modes [PRB 56, 5552 (1997)].

In fact, one of the initial motivations for our experiment was to check for any changes between the phonon spectra across T_c . We have measured the low-energy phonons along both [100] and [001] directions at temperatures above and below the T_c of ~ 3.2 K. However, as we show in Fig. R1, we find that there is essentially no difference for

the results measured below and above T_c . We think there are two reasons for the absence of change below and above T_c : 1), The low T_c of the sample makes the 2Δ too small and difficult to be resolved from the strong Bragg tails; 2), The low carrier density of the sample near the Fermi level makes the strength of the electron-phonon coupling relatively weak. In the revised manuscript, we have added these references and made discussions accordingly.

Fig. R1 **a** and **b**, phonon spectra measured at temperatures above and below the T_c of 3.2 K around the nuclear Bragg peak (1, 0, -5) along [100] and [001] directions, respectively. In **b**, the data at different q s are offset vertically for visibility purpose. The dashed lines denote the value for twice of the superconducting gap.

Comments:

(b) Are the [100] axis (along which the phonon dispersion was taken) and the nematic superconducting axis (one of the [100] axes where H_{c2} exhibit maximum) common or not? If the authors had measured the dispersion both along the [100] and [010], comment whether nematicity is found in the phonon dispersion or not.

Response:

In the neutron scattering experiment, [100] and [010] axes are symmetrically identical, and we did not distinguish between these two axes in the neutron experiment. For the

transport measurement, we believe that the external magnetic field induces a spontaneous symmetry breaking along one of these axes and makes them inequivalent.

Comments:

(c) The explanation in the figure captions such as "integrated with $K = [-0.1, 0.1]$ rlu and $L = [-16, -14]$ rlu" is not clear to non-specialists. Some more explanation should be added about such integration process.

Response:

We have explained the integration process in more details as suggested by the reviewer.

Comments:

(d) What are the white regions extending diagonally in Fig. 2b and Supple. Fig. 2b? I guess that data is missing because of the configuration of the spectrometer. But please explain.

Response:

As the reviewer correctly pointed out, the white streaks are indeed resulted from data missing due to the lack of detector coverage wherein. We have explained this in the caption.

Comments:

(e) In Fig. 3, the data are offset vertically. Explain how much offset is added.

Response:

The main message we want to deliver from Fig. 3 is the lineshape of the phonons at different q s at different Bragg peaks along both the [100] and [001] directions. The purpose of adding offsets, which varies from panel to panel, is to make the lineshape of the phonons clearly visible. The absolute intensities are not that important here so we did not mention the arbitrary offset. But to not mislead readers for the phonon dispersions, we have removed the dashed lines connecting the peaks and changed the descriptions in the caption.

Comments:

(5) Comments on the presentations of the data.

(a) For Fig. 2, it would be better if the vertical scale is common in the all four panels. i.e. all data should be plotted in the energy range between 0 and 6 meV. In addition, for Fig. 2c and d, the horizontal range should include $q = 0$. If data outside of the present ranges is missing, such regions can be left white.

(b) For Fig. 3c, it is written in the figure caption that square root of intensity is plotted. Then, the label of the vertical axis should be modified.

(c) In Fig. 3d, explain how the dashed curves (extrapolating the spectra for $q = 0.5, 0.8,$ and 0.9 to 5 meV) are drawn. If there is no good reason to estimate the position and shape of the TO peaks even without measurements, these dashed curves should be removed since they can be misleading.

Response:

We thank the reviewer very much for the careful reading and for the useful suggestions on improving the figures. We have taken all these suggestions and made changes accordingly in the revised manuscript.

(6) Other minor comments.

(a) In the "Author Contributions", some authors are not mentioned.

Response:

We thank the reviewer again for reading the whole paper so carefully. We have mentioned all the authors' names in the revised version.

Response to Reviewer #2

Comments:

This is a very interesting paper that should be published after the authors have addressed the following minor points:

Response:

We thank the reviewer very much for the positive comment on the paper. We also thank the reviewers for carefully reading the paper and for providing constructive suggestions for us to improve the work. As we elaborate from the point-by-point response below, we have addressed all these issues and made changes accordingly in the revised manuscript.

Comments:

1) The in-plane electrical, thermal, NMR and penetration depth studies indicate a two-fold symmetry rather than the 3 fold symmetry of the crystal structure. This nematic behavior is consistent with p wave superconductivity but may not be definitive test. Do the authors have any knowledge of how the Sr dopants are spatially arranged between the layers?

Response:

In this work, we provide inelastic neutron scattering results showing that there exist highly anisotropic phonons, with the linewidths of the acoustic phonons increasing substantially at long wavelengths, but only for those along the [001] direction. We consider this as strong evidence for the singular-phonon-induced *p*-wave nematic superconducting pairing in this sample. Combining the theory and experimental results from various techniques, we believe it to be a very compelling case.

In the previous scanning tunneling microscopy (STM) measurements on our samples [Nature Communications 8, 14466 (2017)], it is found that the Sr dopants are intercalated between the quintuple layers consisting of Bi and Se. For the samples with Sr in the Bi sites, they are not superconducting. The schematic structure for $\text{Sr}_x\text{Bi}_2\text{Se}_3$ is shown in the way that Sr dopants reside in the van de Waals gaps. In the caption, we state that “The dopant Sr ions are believed to be intercalated in the van der Waals gaps between the quintuple layers consisting of Bi and Se ions. [12]” We now have added a remark in the main text on this issue.

2.) In Fig 2a, why is the TA mode so difficult to see compared with Fig 2a in the supplementary information?

Response:

Such a difference may be largely due to their different sample masses, as they were

measured on the same spectrometer with the same experimental setup. The mass for Bi_2Se_3 and $\text{Sr}_{0.1}\text{Bi}_2\text{Se}_3$ samples used for the neutron experiment were 1.6 g and 1.1 g, respectively. We now include the mass of the Bi_2Se_3 sample in the Methods section and caption in the Supplementary Information so that one will have a better idea on why the intensities are different.

3.) On page 3 , right column, line 4- it should read "remarkable proximity of all pairing channels." It would be helpful if this statement were expanded a bit using the results from ref 23, since this is the key theoretical result that led to the present experiments.

Response:

We thank the reviewer for catching the typo and for the useful comment. Reviewer #1 also has one similar comment requesting more discussions on how the singular phonons induce the nematic superconductivity. We now discuss more on this issue following the suggestions from both reviewers.

4.) Supplementary Fig Caption 1: The last statement that a shielding fraction of 95% indicates a large superconducting fraction is not always true. Filamentary superconductivity can produce a similar result. A heat capacity feature near T_c or a large Meisner fraction are better indicators of bulk superconductivity.

Response:

We totally agree with the reviewer. The large shielding volume fraction is just a possible indication of the large superconducting volume fraction. Nevertheless, these results are consistent with previous STM results on our samples [Nature Communications 8, 14466 (2017)], which show that they have a large superconducting volume fraction. We have now change the sentence as "A ratio over 95% suggests a large superconducting volume fraction in the sample, consistent with previous scanning tunneling microscopy results. [1]"

5) The paper by Willa et al. on the calorimetric evidence for nematic sc in $\text{Sr}_{0.1}\text{Bi}_2\text{Se}_3$ should be cited. (PRB 98, 184509 (2018)).

Response:

This paper is now cited as Ref. 29.

REVIEWERS' COMMENTS:

Reviewer #1 (Remarks to the Author):

(0) I have reviewed the manuscript NCOMMS-19-04391B "Evidence for singular-phonon-induced nematic superconductivity in a topological superconductor candidate Sr_{0.1}Bi₂Se₃" re-submitted for publication in Nature Communications.

In the resubmission letter, the authors addressed all of my and the Reviewer #2's comments almost satisfactory. The manuscript has been revised accordingly with much improvement from the previous version. With these, I recommend the manuscript for publication, after a few minor points listed below are taken into account.

(1) In the authors' response to my comment (4)-b, I also wanted to know whether the [100] axis in INS and the [100] axis in the H_{c2} measurement is common or not. Or, in other words, does the INS [100] axis correspond to 0-deg of Supple Fig.3, or 120-deg, or 240-deg, or not known? Add one comment on this in the text.

(2) I prefer to a paragraph break between "... Ref.37." and "We have also examined the phonons at T=0.8 K..." in page 2, since the latter part discusses a different topic from the topic in the former.

(3) In relation to my comment (4)-d, there are strange dark-red parts in Fig.2(b) and Supple. Fig. 2(b). I guess the color of these regions should be white.

Reviewer #2 (Remarks to the Author):

The authors have adequately addressed my concerns, and I have no objection to the publication of the paper in its present form.

Response to the Reviewers

We thank both reviewers for their efforts in reviewing our manuscript and for the recommendations in publishing our work. We have taken all of Reviewer #1's suggestions into consideration and modified the manuscript accordingly. The changes are marked in red in the manuscript.

Response to Reviewer #1

Comments:

(0) I have reviewed the manuscript NCOMMS-19-04391B "Evidence for singular-phonon-induced nematic superconductivity in a topological superconductor candidate $\text{Sr}_{0.1}\text{Bi}_2\text{Se}_3$ " re-submitted for publication in Nature Communications.

In the resubmission letter, the authors addressed all of my and the Reviewer #2's comments almost satisfactory. The manuscript has been revised accordingly with much improvement from the previous version. With these, I recommend the manuscript for publication, after a few minor points listed below are taken into account.

Response:

We are glad that the reviewer is satisfied with our response. In our response below, we have addressed these issues and made changes accordingly in the revised manuscript

(1) In the authors' response to my comment (4)-b, I also wanted to know whether the [100] axis in INS and the [100] axis in the H_{c2} measurement is common or not. Or, in other words, does the INS [100] axis correspond to 0-deg of Supple Fig.3, or 120-deg, or 240-deg, or not known? Add one comment on this in the text.

Response:

The single crystals for the inelastic neutron scattering and H_{c2} measurements were two different pieces cut from the same batch of the as-grown crystal. During each measurement, we chose one of the three [100] directions independently. A comment has been added in the caption of Supplementary Fig. 3.

(2) I prefer to a paragraph break between "... Ref.37." and "We have also examined the phonons at $T=0.8$ K..." in page 2, since the latter part discusses a different topic from the topic in the former.

Response:

Now they are divided into two paragraphs.

(3) In relation to my comment (4)-d, there are strange dark-red parts in Fig.2(b) and Supple. Fig. 2(b). I guess the color of these regions should be white.

Response:

The dark red areas indeed are due to the lack of detector coverage. These areas have now been plotted in white in Fig. 2 and Supplementary Fig. 2.

Response to Reviewer #2

Reviewer #2 (Remarks to the Author):

The authors have adequately addressed my concerns, and I have no objection to the publication of the paper in its present form.

Response:

We are glad that the reviewer is satisfied with our response. We thank the reviewer very much for the efforts in reviewing this work.